# LimbNET: collaborative platform for simulating spatial patterns of gene networks in limb development

Antoni Matyjaszkiewicz [1,2]✉ & James Sharpe [1,2,3]✉

## Abstract

**Successful computational modelling of complex biological phenomena will depend on the seamless sharing of models and hypotheses between researchers of all backgrounds—experimental and theoretical. LimbNET, a new online tool for modelling, simulating and visualising spatiotemporal patterning in limb development, aims to facilitate this process within the limb development community. LimbNET enables remote users to define and simulate arbitrary gene regulatory network (GRN) models of 2D spatiotemporal developmental patterning processes. Researchers can test and compare each others' hypotheses within a common framework. A database of previously created models empowers users to simulate, explore, and extend each others' work. Spatiotemporally varying gene expression intensities, derived from image-based data, are mapped into a standardised computational description of limb growth, integrated within our modelling framework. This enables direct comparison not only between datasets but between data and simulation outputs, closing the feedback loop between experiments and simulation via parameter optimisation. All functionality is accessible through a web browser ([https://limbnet.embl.es](https://limbnet.embl.es)), requiring no special software, and opening the field of image-driven modelling to the full scientific community.**

**Keywords** Computational Modelling; Gene Regulatory Networks; Limb Development; Online Community Platform; Pattern Formation
**Subject Categories** Computational Biology; Development

## Introduction

A key challenge for the future of computational modelling is how to easily share models, simulations, and hypotheses between a diverse community of theoreticians and experimentalists. This is essential so that different researchers can test and compare each other's hypotheses and to allow their new results to build on previous

discoveries. In this publication, we present LimbNET, a new accessible online simulation tool for intuitive image-driven computational modelling, simulation and visualisation of gene expression patterns important to limb development, and the sharing of associated results within the community.

At the heart of biology is the concept of mechanism. For some biological questions, this may be a molecular interaction, but to explain developmental pattern formation, gene regulatory networks (GRNs) are the appropriate causal approach (Jaeger and Sharpe, 2014). By creating and comparing mechanistic hypotheses and models (Sharpe, 2017) of how something works, we may infer the underlying behaviour of complex biological systems in various model systems, e.g. Drosophila development (Jaeger et al, 2004b), neural tube (Balaskas et al, 2012) and limb development (Raspopovic et al, 2014; Uzkudun et al, 2015; Onimaru et al, 2016). While data and analysis tools are increasingly shared online, competing hypotheses are still largely the preserve of each individual lab. With LimbNET, we propose a new approach: the sharing of hypotheses formalised as mathematical models and concrete computational simulations, integrated with associated experimental data through a user-friendly common framework. In this respect, LimbNET's core goals are:

- Sharing and accessibility,
- Standardisation of both models/hypotheses and data via a common platform: an atlas of 2D spatiotemporal gene expression patterns and a repository of models/simulations,
- Reducing the "energy barrier" of the modelling process, especially for non-experts.

More precisely, our desired system should facilitate investigation of novel signalling networks, integrate models with related data, which is also hosted on the same platform (within a common frame of reference), permit online simulation, and collation of models and corresponding inputs/outputs into a unified database that permits model comparison. It should facilitate the exploration of hypotheses—and thus models—both published and unpublished. It should be easy for a scientist to explore an existing model or, indeed, to challenge it. The platform should support not only tweaking and exploration of models but also more rigorous parameter fitting and

[1]European Molecular Biology Laboratory, EMBL Barcelona, C/ del Dr. Aiguader 88, PRBB Building, Barcelona 08003, Spain. [2]Barcelona Collaboratorium for Modelling and Predictive Biology, Carrer de Wellington 30, Barcelona 08005, Spain. [3]Institució Catalana de Recerca i Estudis Avançats (ICREA), Passeig Lluís Companys 23, Barcelona 08010, Spain. ✉E-mail: antoni.matyjaszkiewicz@embl.es; james.sharpe@embl.es

optimisation of models onto existing experimental data (Mousavi and Lobo, 2024; Uzkudun et al, 2015).

With LimbNET, we hope to implement the above goals by addressing the following challenges:

Firstly, within the community of researchers who study limb development, scientists and labs are globally geographically dispersed, and there is a large number of genes, pathways and mechanisms to investigate, necessitating a coordinated approach. Communication is generally indirect and asynchronous (research papers and scientific conferences). In order to integrate and coordinate data and ideas across a geographically diverse research community, LimbNET's primary interface consists of an accessible web client, central database, and a common server platform. There is no need for scientists to install software, build further modelling tools or compile libraries in order to collaborate on each other's models (Byrne et al, 2010). In the last decade, the trend towards online collaboration and web applications has only accelerated; this ongoing proliferation of online platforms has led to a gradual shift in the philosophy of data management, analysis, and modelling from predominantly local, to cloud-based. Precipitated by online databases (as opposed to physical media (Emmert et al, 1994)), the modern web-based science ecosystem now encompasses not only visual analysis tools (Lyons et al, 2022) but also online modelling and/or simulation (Fortmann-Roe, 2014; Wortel and Textor, 2021), with established platforms such as Netlogo (Tisue and Wilensky, 2004) also providing online versions.

Secondly, not only are hypotheses and models geographically spread, but also the data. Open online distribution of data has long been recognised as crucial to accelerating bioscience (Emmert et al, 1994; Baxevanis and Bateman, 2015). While publications disseminate results, this does not always guarantee or ensure access to data. Although freedom of—and ease of access to—experimental data is rapidly increasing, even in the case that data are shared, non-standardised interchange formats may complicate re-use. LimbNET's centralised database of gene expression patterns from the limb development community aims to rectify this by providing the data in such a way that the modelling and simulation tools can directly access it, with no concerns about file formats or matching protocols.

The field of limb development is rich with data; many aspects of limb development are well-characterised, and it is a paradigm model system of the more generalised problem of developmental patterning and organogenesis (Petit et al, 2017; Capdevila and Belmonte, 2001). The limb bud exhibits a rich variety of dynamical patterning behaviours during development (Benazet and Zeller, 2009; Raspopovic et al, 2014). Of these, the vast majority of characterised behaviours have been described primarily experimentally. The developmental trajectory of the limb bud consists of a well-defined time course, and so data can be accurately staged and mapped into a consistent common reference framework (Musy et al, 2018). Within LimbNET, we provide tools for staging and digitisation that are tied into the database and model framework. This allows provenance to be tracked, back to the original imaging data which can finally be used to fit model parameters, closing the modelling loop.

The final challenge—and the hardest—is that of mixed expertise throughout the field. Existing modelling and simulation frameworks tend to be generalised, with the aim of being able to model many different systems (Mirams et al, 2013; Starruß et al, 2014), but thus requiring significant effort and domain-specific knowledge in order to create a model of a specific organ. Some classes of models can also be shared through centralised repositories such as BioModels (Malik-Sheriff et al, 2020), or as part of the community for a specific tool, as in the case of Morpheus (Starruß et al, 2014). However, simulation and modelling are often limited to a small group of specialised "theoreticians", while our goal is to bring experimental researchers into this endeavour. While researchers may not be familiar with the details of implementing a mathematical model, they should nonetheless be able to leverage an existing framework to simulate their hypotheses, thus concretising abstract ideas. Our core goal is to be able to rapidly gain intuition of a given system's behaviour, to tighten the iterative loop (Harline et al, 2021) of model design, building and testing. Indeed, while it is common to study genes one by one, disentangling the complex causal relationships of a system's network dynamics requires a systems biology approach. By providing a straightforward interface for the definition and simulation of models, we hope to lower the energy barrier for model development and streamline the modelling process. Our repository of models/simulations—including examples, tutorials and published hypotheses—will facilitate creation of models, either de novo or by extending/modifying an existing model/hypothesis, leading to a truly integrated and collaborative modelling process in which scientists build on others' work, while testing and comparing existing hypotheses. Finally, we hope that such a system may provide an improved ontology, a platform upon which to build standardisation of limb development terminology (e.g., a controlled and well-specified coordinate system).

Only a small proportion of publications in the field contain a strong computational modelling perspective, thus there exists a large quantity of 2D gene expression data, published over the last decades that has not been used in mathematical model development. Clearly, although formal modelling has not been performed in these cases, a wide range of tested hypotheses are associated with these data, predicting interactions that lead to patterning behaviours. However, we believe that by formalising these hypotheses as mathematical models, we can gain further insight into the predicted mechanisms. The collection of researchers studying limb development is a tight-knit community, and therefore creating a tool that can be used by the whole community is a tractable goal.

It is important to note that while many generalised modelling platforms already exist, general-purpose simulation tools may sometimes cover too broad a scope, requiring significant work to build up a realistic model of a specific organ. By contrast, LimbNET is consciously tailored towards a specific organ—the developing limb—so that researchers who specialise in this particular model system have a lower energy barrier to start simulating and exploring their hypotheses.

## Results

LimbNET's design follows a client-server architecture (Fig. 1). There is a direct logical separation of the user-facing interface, referred to as the LimbNET client (Fig. 1A), and the underlying implementation of models (Fig. 1B) and simulations/data storage, referred to as the server or backend (Fig. 1C). For the user—typically an experimental scientist —this has numerous advantages: for instance there is no need to deal with problems of data representation, storage, or computation, all of which are dealt with remotely. The web-based client further simplifies

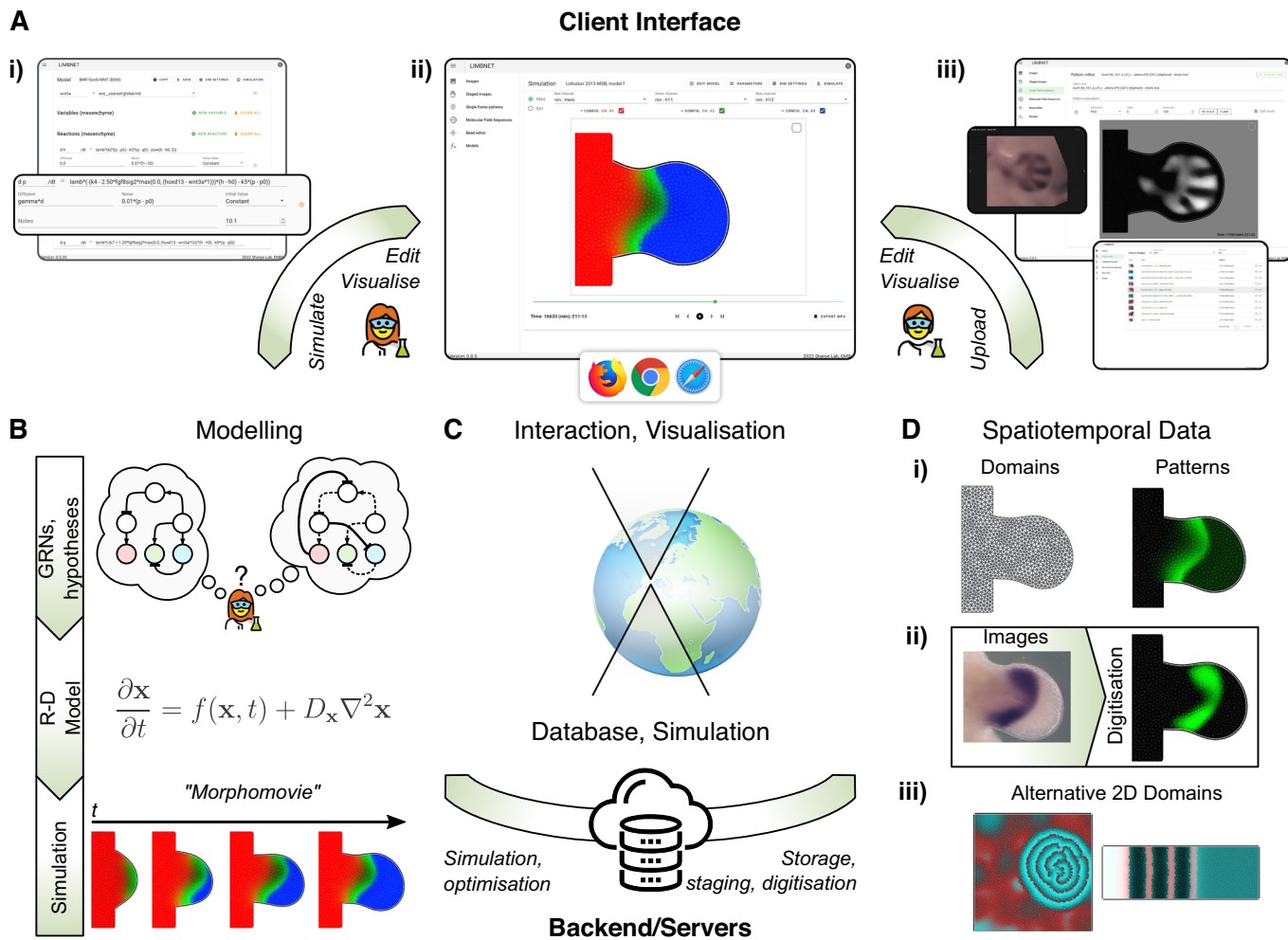

**Figure 1. High-level schematic overview of the LimbNET platform.**

(**A**) LimbNET's user-facing client is a single-page web application, a unified GUI exposing specific views corresponding to defined modelling or data visualisation tasks: (i) model definition; (ii) visualisation of model simulation; (iii) upload/visualisation of image-based spatiotemporal data, and corresponding digitisation. (**B**) LimbNET facilitates a streamlined workflow for modelling, simulating and comparing patterning hypotheses. Users' input GRNs represented by spatial patterning GRN equations ((A,i), above). LimbNET simulates the model, outputting "morphomovies" that illustrate simulated spatiotemporal patterns over a dynamic 2D domain. (**C**) The client GUI communicates with the backend (databases, model simulation engine). Users need not deal with the computational requirements of heavy numerical simulation, nor the associated storage of results, which are streamed to the web client as required. (**D**) LimbNET specialises in 2D spatiotemporal data, capturing time-dependent scalar quantities (gene expression, ligand concentration, etc.) over a dynamic 2D domain. (i) Spatiotemporal patterning data are stored internally along with the relevant domain shape(s)—e.g., mouse embryonic hindlimb bud - and their corresponding discretisation (a triangular mesh). (ii) Uploaded image-based data (e.g., in situ hybridisation gene expression), stored in our database, can be digitised (in situ image in this panel reproduced from (Uzkudun et al, 2015)). Digitisation maps image intensity onto a discretised 2D domain. These patterns can then be used in models and simulations in LimbNET. (iii) Versatility—although LimbNET is tailored towards limb development, it can also provide alternative pre-defined growing or static meshed domains. (Left) a snapshot of the classic "Brusselator" (Prigogine and Lefever, 1968) on a square domain; (right) the Progressive Oscillatory Reaction–Diffusion (PORD (Cotterell et al, 2015)) model in a rectangular domain.

access. Non-specialist users can access all functionality—data visualisation, modelling, and simulation (Fig. 1A)—through the web browser without the need for installation (let alone compilation from source).

All relevant functionality is presented as a user-friendly Graphical User Interface (GUI), with no need for prior programming knowledge. It is worth noting that this form of interaction may be disadvantageous for power users; however should the need arise, for example, for automation purposes, an Application Programming Interface (API) is also available which may be accessed programmatically via external scripts and tools. LimbNET

has been written using standard technologies that are in widespread use throughout the software industry, and have been proven to behave robustly. Documentation is provided online specifying common usage and tasks, tutorials and examples, and implementation details.

## Client

LimbNET's client, the core of the user-facing experience, is written in a mixture of javascript and typescript and implemented as a single-page web application (SPA), primarily using the Vue.js

framework (http://vuejs.org/). The SPA format presents the user with a single interface, similar to a desktop application, with multiple "views" corresponding to different tasks the user may want to perform (Fig. 1A): for example, definition of a model specification and corresponding equations (Fig. 1Ai); visualisation of spatiotemporal patterns either arising from digitised data or from model simulation outputs (Fig. 1Aii); visualisation, editing and digitisation of experimentally acquired imaging data (Fig. 1Aiii).

The application offers task- or data-oriented views, in a hierarchy of searchable and filterable listings for each data type. Specific views for each data type allow users to view/visualise a given dataset, and perform specific data-oriented tasks such as editing related metadata or using data for further experiments. For a given type of data, the owner may decide whether to make it public (available to the whole LimbNET community) or keep it private (viewable only by the owner).

## Modelling

The core objective of LimbNET is to facilitate the straightforward modelling, simulation and comparison of hypotheses for a given patterning mechanism (Fig. 1B). Users input their desired GRN (Fig. 2A) to LimbNET as a model specification (Fig. 2C). Or indeed, users can simply browse and run models which are already uploaded into the system. For example, the model shown in Fig. 2 can be found in the web client's model menu with the name "Uzkudun et al, 2015, model C". Models represent high-level abstractions of patterning, defined by dynamical systems of coupled reaction–diffusion partial differential equations (PDEs) (Fig. 2B). Simulation of a given model produces a single compound output—a "morphomovie"—a collection of spatiotemporal patterns, over a moving and growing 2D domain (Fig. 2D). Simulations are thus concrete realisations of a model—a given spatiotemporal patterning process—arising from numerical simulations of a model's equations under specific parameter sets and virtual experimental conditions.

Users can browse, filter and view all publicly available models and corresponding simulation results, including those from previously published works, tutorials, examples/case studies, and user-shared public models. Currently, models from three previous publications are available (Raspopovic et al, 2014; Uzkudun et al, 2015; Onimaru et al, 2016), but more will be made available over time. Users may create de novo model specifications, and also copy or fork existing models, in order to further edit them and build upon previous work. Through the model editor (Fig. 2C) users may create and freely edit model specifications, and define global parameters, variables, and PDEs. All relevant editable components of a model specification are exposed as freely editable building blocks of the corresponding PDE system. Global parameters and variables can be defined, along with expressions for their respective values. Potential errors and model inconsistencies are highlighted in the GUI.

The core limb bud model consists of two compartments: the mesenchyme (a 2D domain on which the majority of patterning occurs) and an optional ectoderm (the 1D linear boundary of the limb bud). Within compartments, users may freely define and tailor variables and reactions. PDEs are specified according to their derivatives' right-hand-side, along with expressions for diffusion terms, noise and initial conditions. Moreover, users can choose from various geometries for the model/simulation domain (Fig. 1Diii).

In addition to standard variables/reactions, users can specify dynamic pre-defined spatiotemporal pattern variables that evolve over time independently of other model entities. These represent temporal sequences of digitised images, mimicking experimental data for direct comparison of model outputs to experimental observations. Secondly, they can be chosen to act as feedforward inputs to the rest of the system, receiving no feedback from other variables. For example, in the model shown in Fig. 2, the region where Fgfs are expressed is considered a pre-defined input to the system, labelled as fgf4_in and fgf8_in. This permits auxiliary functionality such as pre-computing morphogen patterns known to act only as one-way inputs, or defining other experimental interventions such as mobile ligand-soaked beads (Mercader et al, 2000).

Once a model is defined and saved, users may initiate simulations under a given domain shape, a set of parameters and a set of inputs. The platform performs the numerical simulation for the given model specification remotely on the server—this way, the efficiency of the calculation does not depend on the user's own computer. The results of the simulation are a set of spatiotemporal patterns which are automatically transferred to the client browser for browsing, comparison and visualisation in the simulation view (Fig. 1Aii).

In the simulation view, the user may examine the 2D domain and analyse all the spatiotemporal patterns (3 may be seen at any one time represented by red, green and blue channels in the visualisation). Crucially, the user has the flexibility to freely slide back and forth through time, while adjusting the intensity of each channel independently to visualise the temporal dynamics of their system within the growing and deforming mesh which represents the tissue movements of the limb bud. For each of the three channels, the user is free to select between model outputs and digitised spatiotemporal data, thus the model output can be directly compared to digitised experimental data in a unified interface. In order to further facilitate comparison between simulated and experimental data, we also provide two views specifically structured for data comparison. First, a side-by-side double-viewer mode, so researchers can (for example) compare real digitised data with the simulated results. Second, an image subtraction mode, as a direct way to visualise the differences between any two patterns (differences between simulated and real patterns, or two different simulated patterns, or two real gene expression patterns).

## Image-based data

The simulations create predictions of dynamic gene expression patterns over time (as well as other molecules, such as the retinoic acid in Model C above). But in addition to predicting patterns, it is important to be able to upload known gene expression patterns from data, which generally means image data of in situ hybridisations.

In LimbNET, users are able to upload and view single-channel images of gene expression patterns in limb buds. Images are generic and may represent qualitative or quantitative acquisition (Fig. 1D), including whole-mount in situ hybridisation, fluorescence microscopy, in situ hybridisation chain reaction (HCR), or other techniques. Our image database can be searched and filtered to find expression patterns related to a specific gene or experimental

**A**

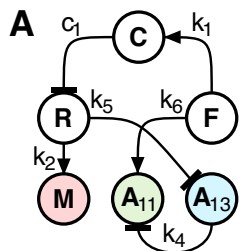

**B**

$$\frac{\partial F}{\partial t} = P_F - \lambda_F F + D_F \nabla^2 F \qquad \frac{\partial M}{\partial t} = P_M \frac{R^\mu}{R^\mu + k_2{}^\mu} - \lambda_M M$$

$$\frac{\partial R}{\partial t} = P_R - c_1 CR - \lambda_R R + D_R \nabla^2 R \quad \frac{\partial A_{11}}{\partial t} = P_{A_{11}} \frac{F^\mu}{F^\mu + k_6{}^\mu} \frac{k_4{}^{\mu'}}{A_{13}^{\mu'} + k_4{}^{\mu'}} - \lambda_{A_{11}} A_{11}$$

$$\frac{\partial C}{\partial t} = P_C \frac{F^\mu}{F^\mu + k_1{}^\mu} - \lambda_C C \qquad \frac{\partial A_{13}}{\partial t} = P_{A_{13}} \frac{k_5{}^\mu}{R^\mu + k_5{}^\mu} - \lambda_{A_{13}} A_{13}$$

**C**

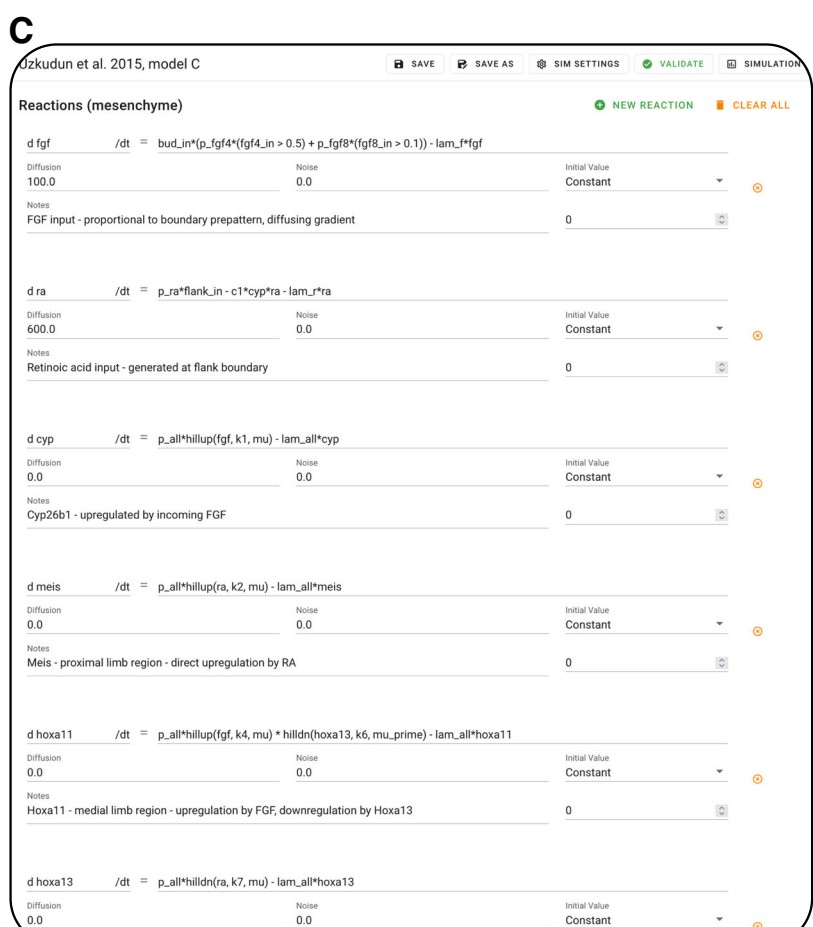

**D**

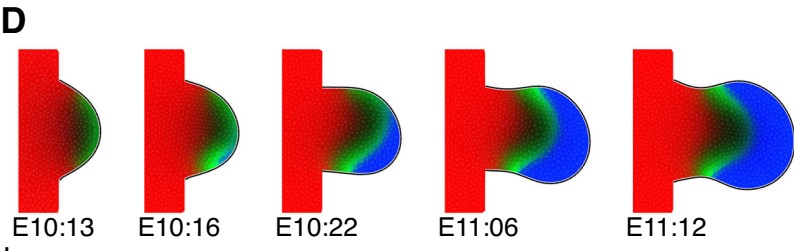

E10:13    E10:16    E10:22    E11:06    E11:12

t ⟶

**Figure 2. Limb patterning network model specification and simulation.**

(A) One way of describing patterning processes is as a network of interactions— mutual regulation of "morphogens", genes, metabolites and other relevant spatiotemporal quantities. The network shown describes a possible mechanism for proximodistal patterning in the limb bud ("Model C", taken from (Uzkudun et al, 2015)); Variables are defined as follows: F, FGF; R, retinoic acid; C, Cyp26b1; M, Meis; A11, Hoxa11; A13, Hoxa13. (B) The system of coupled Reaction–Diffusion PDEs corresponding to the network in (A). (C) LimbNET's client-side model editing interface, showing the equations corresponding to the system in (B), as entered by the user, as well as initial conditions, diffusion, and noise, where relevant. In addition, the editor allows users to create named variables, global parameters, and pre-defined patterns. Many models such as this one are provided in LimbNET's model menu, so users do not have to type these from scratch. (D) LimbNET's predicted patterns for Meis (red), Hoxa11 (green) and Hoxa13 (blue) genes, resulting from numerical simulation of the model defined in previous panels.

methodology, for example. Images uploaded and digitised by a user may be kept private for the user, or made public to the community.

### Staging and digitisation

Image processing functionalities are also provided to users within the web-based platform. LimbNET's image viewer is integrated with our quantitative Embryonic Mouse Ontogenetic Staging System (Musy et al, 2018). Given an image of an embryonic limb the user draws an approximate outline and the system will return the developmental stage of the embryo, based on the morphometry of that limb bud (generally to an error of $+/-$ 1 h), as well as the canonical outline shape for a limb of that developmental stage (Fig. 1Di).

Staging an image maps it to a canonical limb bud shape and the representation for that given developmental stage. Intensity of a spatial pattern (e.g., gene expression) within the image may then be "digitised", i.e., mapped onto the canonical domain shape (Fig. 1Dii). A series of digitised patterns, corresponding to ordered time points, may be combined into sequences which may then be used in the modelling and simulation pipeline.

## Server/backend

LimbNET's backend—server components—provides the web API, database and all associated computational capabilities needed by the web client. The backend consists of a HTTP server, API server, database, task queues and a computational server. This implementation separates web service functionality—direct communication with clients—from longer-running asynchronous tasks (analysis and computation).

The LimbNET API server is written in Python using the Flask (http://flask.palletsprojects.com) framework, and the database is implemented using PostgreSQL (http://www.postgresql.org). In order to separate web services from longer-running asynchronous jobs we implemented a task queue using Redis (http://redis.io/) and RQ (http://python-rq.org/). Aside from benefiting from a separation of concerns, this makes the simulation system scalable should the need arise: computational tasks can be distributed as we see fit across one or more remote workers, for example, during parameter optimisation.

Computational tasks consist of two main categories: analysis of image-based data, and simulation. Image analysis includes staging (as discussed above) which uses the system published in (Musy et al, 2018). Associated digitisation and offline graphical tasks are handled using the Vedo library (Musy et al, 2024). Simulation tasks consist of a hierarchy of job dependencies. Firstly, the model specification is validated and parsed. The parsed model specification is then used to generate an internal model representation for our custom simulation engine written in c++. This is the final

artefact required to run a simulation job: this task is dispatched to a remote worker, which runs the simulation, reports progress to the client, and finally returns a computed result that is stored with the associated model and can be further analysed by the user.

## Simulation

### Reaction–diffusion expressions

Simulation in LimbNET consists of the numerical integration of a set of spatial patterning GRN PDEs by our simulation engine, according to a user-supplied model specification (Fig. 2). A network of interactions (Fig. 2A) represents the interplay of regulatory components that contribute to a patterning process. Our model specification structure represents a system of coupled (stochastic) PDEs (Fig. 2B), modelling interactions in time and space over a growing 2D domain. The equations may be expressed in the general form

$$\frac{\partial \mathbf{u}}{\partial t} = f(\mathbf{u}, \mathbf{x}, t) + \mathbf{D}\nabla^2 \mathbf{u}, + \eta(\mathbf{u}, t)$$

where $\mathbf{u}$ is a vector of all reactants in the system, $t$ is time, $\mathbf{x}$ is our spatial coordinate, $\mathbf{D}$ is a diagonal matrix of all per-species diffusivities and $\eta$ is a vector of all per-species noise terms.

LimbNET's principal model implements a dynamic 2D domain representing a growing limb bud, which is an extension of the model published in (Marcon et al, 2011). The model comprises three compartments: the 2D limb domain (mesenchyme), its 1D boundary (ectoderm), and a non-spatial global compartment. Users may define named parameters/numerical constants as well as time-dependent variables and reactants, either globally or in a spatial compartment.

Reactants in both the ectoderm and the mesenchyme are user-defined equations expressing the reaction component in the right-hand-side of a spatiotemporal derivative - $f(\mathbf{u}, \mathbf{x}, t)$ in the system above (Fig. 2C). They may be explicit functions of time, of global entities, and of entities in the same compartment. They are (optionally) subject to diffusion and noise, and are explicitly integrated in time. Diffusion and noise coefficients may additionally be defined as spatiotemporally varying expressions of the same form as compartmental variables.

Compartmental variables are similar to reactants. However—unlike reactants—they are not the right-hand side of a derivative, and thus are not integrated in time, neither are they able to diffuse or to be affected by noise. Both variables and reactants have explicit spatial dimensionality corresponding to their compartment, and connections between the mesenchyme and ectoderm compartments

are defined in terms of a diffusive coupling between a user-selected pair of reactants.

Furthermore, pre-defined patterns can be created within each compartment. These may change in space and time, but are fixed with respect to the rest of the model, acting only as feedforward inputs to the system.

The user may specify as many entities as desired (within practical limits; too many variables may have a prohibitive impact on computation time). The client GUI, exposes the right-hand side(s) of the above PDEs as editable fields (Fig. 2B), such that $f(\mathbf{u}, \mathbf{x}, t)$, $\mathbf{D}$, and $\eta$ can all be separately written as free mathematical expressions, functions of other user-defined entities in the model. A selection of standard mathematical functions are available for use in all expressions, as well as pre-defined functions useful for constructing models of GRNs (e.g., Hill-type up-/down-regulation of a reactant). It is up to the user how they define their model: there is no specific restriction to GRNs, nor do variables and reactants in the system need to represent genes; a variable or reactant simply represents any spatiotemporally varying scalar quantity.

### Simulation domain and spatial discretisation

LimbNET's simulation engine is designed to model dynamical patterning processes in 2D during early embryonic limb development. The boundary and 2D geometry of the simulation domain is dynamic over time, representing the known tissue movements during limb development. The standard wild-type growth has been pre-specified, based on analysis of clonal experiments (Marcon et al, 2011)—it does not dynamically respond to control from the GRN. While this might be seen as a limitation to modelling certain limb development phenotypes, the current focus of LimbNET is to address the complexities of GRN dynamics per se. A large proportion of reported perturbations (genetic and also non-genetic, such as bead experiments) reveal shifts in gene expression patterns without significant growth defects. Indeed, the most powerful perturbation results for the task of deciphering GRNs are this large number which primarily alter molecular patterning. In the future, LimbNET will be extended to address growth defects as well, a topic that is addressed further in the discussion.

Using a standard wild-type description of growth also facilitates comparison between models, data, and patterning in general, since it ensures that all spatiotemporal patterns share a common discretised domain. In addition, since meshing a discretised domain is only precomputed once, the overall speed of computation is improved, especially when it may be necessary to run hundreds of simulations (for example, during parameter optimisation). Growth, stretching, and changes in shape are essential to our model, so ensuring the cost of remeshing is only paid once in advance is valuable in order to maintain good performance.

Creating the pre-defined dynamical mesh was done as follows. First the boundary was defined, in this case a curve defining the canonical mouse hindlimb developmental trajectory (Musy et al, 2018), captured at hourly time points (Fig. 3A). This boundary was combined with a small rectangular region corresponding to the embryonic flank, and the entire domain was then discretized as a mesh composed of triangular elements, representative examples of which can be seen in Fig. 3B. In addition, the boundary itself was discretized using linear elements, where any linear element coincides one-to-one with the neighbouring edge of its corresponding triangular element (Fig. 3C). Thus the inner domain,

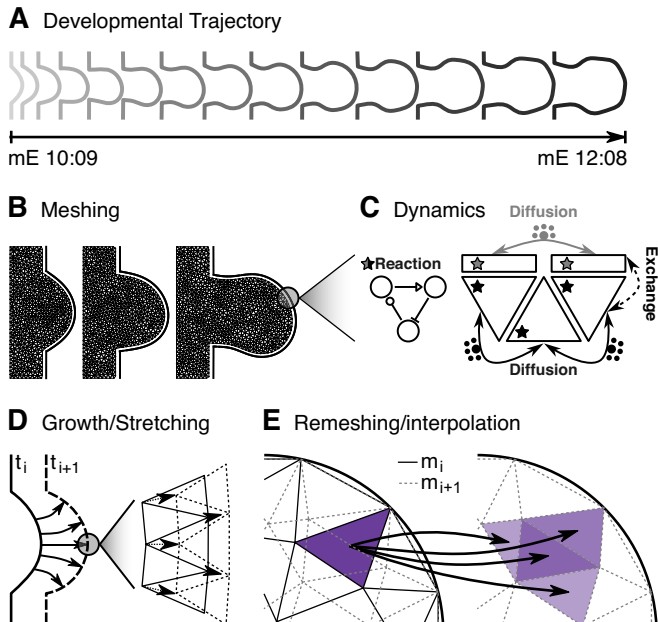

**Figure 3. Embryonic limb bud model dynamics and numerical simulation.**

(A) Selected snapshots of the canonical mouse hindlimb developmental trajectory(Musy et al, 2018) from mouse embryonic stage mE10:09 (10 days and 9 h), through to mE12:08. Simulated reaction–diffusion processes take place on a 2D domain bounded by this canonical shape trajectory. (B) The limb bud boundary (A), with the addition of a rectangular domain representing the flank, is meshed with triangular elements at hourly intervals to form the discretised mesenchyme domain(s) used for subsequent numerical simulations. Three representative meshes are shown; the full trajectory comprises 48 meshes, one per hour of the two-day developmental trajectory. The limb bud boundary (ectoderm; *solid line*) is separately discretised with linear elements. (C) Numerical simulation of a model's GRN reactions. For a single time step of a given simulation, diffusion is integrated according to a finite-volume scheme. In the mesenchyme, diffusion occurs only between neighbouring triangular elements (black arrows); in the ectodermal boundary, diffusion occurs only between neighbouring linear elements (grey arrows). Material can be exchanged between the ectoderm and the mesenchyme (dashed line). At every time step, reaction proceeds according to the numerical solution of the system of PDEs corresponding to all given GRN species, integrated per individual element (*stars*) in mesenchymal and/or ectodermal compartments as relevant. (D) Within each hourly interval, a single mesh is defined, however growth and stretching of the simulation domain is treated as a smooth and continuous process. Thus within a given hour of simulated development, the meshed domain (inset) deforms such that its original shape (solid lines)—derived from the canonical limb bud boundary at the beginning of the hour—fits the boundary at the end of the hour (dashed lines) (Marcon et al, 2011). (E) The mesh is replaced at hourly time intervals, as described in (B). Upon mesh replacement, all per-element scalar quantities (reactants and variables) are interpolated from the mesh at the end of the previous time point ($m_i$, black solid lines) onto the mesh at the current time point ($m_{i+1}$, grey dashed lines). For any given element in the original mesh, a scalar quantity (purple) is interpolated onto elements in the new mesh with which it intersects proportionally according to the ratio of overlapping area between the elements.

triangle-meshed, forms a compartment which effectively represents the limb bud mesenchyme and the outer boundary, formed of linear elements, the corresponding limb bud ectoderm. Material can be exchanged between the mesenchymal and ectodermal compartments, akin to a dynamical boundary condition, which could represent diffusion of a substance between the two compartments.

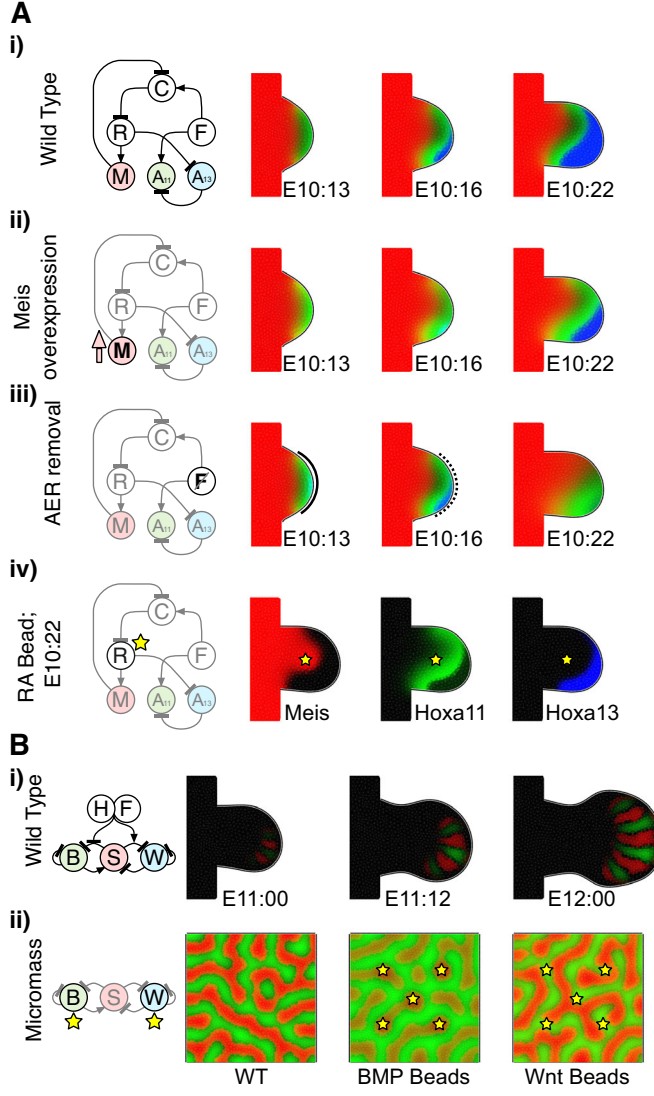

**Figure 4. LimbNET can reproduce existing models.**

An illustrative sample of the variety of patterning mechanisms and experimental interventions that can be modelled using LimbNET. (**A**) Proximodistal (PD) patterning, an implementation of "Model F", the final regulatory network scheme introduced by Uzkudun et al (Uzkudun et al, 2015). Multiple experiments are shown via adjustments to the same model, illustrating the flexibility of LimbNET. (i) The "wild-type" model shows typically observed expression patterns of Meis (red), Hoxa11 (green) and Hoxa13 (blue). (ii) Global Meis overexpression induces a distal shift of the expression boundary between Hoxa11 and Hoxa13 (Mercader et al, 2009). (iii) Simulation of Apical Ectodermal Ridge (AER) removal at E10:16 (Vargesson et al, 2001). (iv) Introduction of beads soaked in retinoic acid induces localised overexpression of Meis and a distal shifting of the Hoxa11 pattern(Mercader et al, 2000). (**B**) The BSW model of digit patterning(Raspopovic et al, 2014), predicts correct number and initiation of digits following a Turing patterning mechanism involving a GRN of Bmp, Sox9 and Wnt signalling. (i) The "wild-type" patterning model simulated in a limb bud domain, reproduces the results of Raspopovic et al, Fig. 3D; Sox9 shown in red, BMP shown in green. (ii) Further experimental interventions can be simulated: left, the network is simulated on a square domain, representing cell culture conditions; *centre*, we introduce BMP-soaked beads to the cell culture, which locally upregulate Sox9 expression; right, we introduce Wnt-soaked beads which locally repress Sox9 while upregulating its expression over a longer range.

Whenever the user performs a simulation, numerical integration is performed at every integration time point to solve the PDEs of the system (Fig. 3C), for both the reaction and diffusion terms. A combination of an explicit Euler-Maruyama (Kloeden and Platen, 1992) scheme for the reaction terms, and a finite-volume scheme for diffusion is used by the solver. Diffusion is integrated between pairs of neighbouring elements of the same type. Finally, where specified, a diffusion-like exchange of material is computed between neighbouring elements of different types (in this case, the boundary elements and those on the interior of the domain). Noise is approximated according to a Wiener process (a one-dimensional Brownian motion) scaled per species. The outputs of the resulting simulation can be viewed within the user interface (Fig. 2D).

Growth and thus a gradual reshaping of the domain boundary necessitate regular remeshing to avoid degenerate or distorted elements. As in previous studies(Marcon et al, 2011; Uzkudun et al, 2015; Raspopovic et al, 2014), we chose to generate a new mesh at every hourly time-point, corresponding to the time points at which new boundaries are defined. Within each hourly stage, the boundary is assumed to grow smoothly (Fig. 3D) and as it grows it induces a stretching and deformation of the corresponding mesh. Finally at the hourly time points where a new mesh has been computed, spatial data from the current time point's mesh is interpolated onto the next (overlapping) mesh proportional to the amount of overlap between elements in the two meshes (Fig. 3E). The stretching within an hourly time interval is defined so that, at the mesh changeover point, the mesh from the previous time interval has deformed such that its boundary overlaps exactly with the new boundary; all material within the domain is conserved.

The simulation framework described here can be generic, and could be applied to other growing and deforming domains (of other organs), but in the LimbNET project, we currently focus on the developing limb bud. Extending the system beyond the limb bud would only require new domains, discretisations, and corresponding data structures. Indeed we already provide a model of a homologous structure in another organism—the developing catshark fin bud (Onimaru et al, 2016))—multiple versions of the mouse limb bud with different discretisations (to validate stability of numerical experiments), as well as domains with more basic geometry (used to approximate patterning in cell culture experiments, for example).

## Case studies

To demonstrate the flexibility of the LimbNET platform in developing different types of models, we implemented two very different previously published models of patterning processes within the embryonic mouse limb bud.

The first model we chose to replicate was the proximodistal (PD) patterning model of Uzkudun et al (Uzkudun et al, 2015), a good example of a GRN model to explain dynamic gene expression patterns. From a high-level point of view, the network (as in Figs. 2A and 4A) consists of two inputs— retinoic acid (RA), and FGF—representing signals from, respectively, the proximal and distal ends of the growing limb bud. The network is largely composed of feedforward interactions with one feedback, and the regulatory interactions comprise both inhibition and upregulation,

modelled using a combination of Hill-type dynamics as well as more general mathematical functions (Fig. 2B,C). The network was implemented in LimbNET and simulated, obtaining results that agree with those previously published (Fig. 4Ai).

To illustrate typical use of LimbNET, we started with this "wild-type" model —depicting experimentally observed behaviours of the system under normal conditions—and we built upon the original equations, incorporating a number of experimental interventions, as was done in the original paper (Uzkudun et al, 2015). Our unified model was expanded to create a single model specification that takes into account three experiments and their observed consequences. Firstly, global overexpression of Meis1 has been shown to induce a distal shift of the expression boundary between Hoxa11 and Hoxa13 (Mercader et al, 2009). This can be simulated by a uniform upregulation of Meis throughout the limb bud (Fig. 4Aii). Secondly, we simulated removal of the AER (Fig. 4Aiii), which can cause a loss of expression of Hoxa13 (Vargesson et al, 2001). This simulation was easily implemented by adding a time-dependent expression into the equation of the production of FGF (Fig. 4Aiii). Finally, implanting a bead soaked in retinoic acid into the middle of the limb bud has a similar impact as the Meis overexpression. It shifts distally the expression boundary between Hoxa11 and Hoxa13 (Mercader et al, 2000). A tool is provided within LimbNET allowing the user to define virtual bead experiments. Both the position and the timing of bead insertion can be defined, by clicking on a given triangle at a particular time-point. This definition can then be linked to the production of any variable in the model—in this case retinoic acid, resulting in a recapitulation of the originally published simulation (Fig. 4Aiv).

The second model we chose is a very different type of GRN, whose spatial patterning dynamic is dependent on feedback loops. It is the proposed BSW model (Raspopovic et al, 2014), a Turing system to explain the patterning of digits through the interactions of Bmp, Sox9 and Wnt signalling. Unlike their more straightforward mechanistic relatives, these equations contain a number of nonlinear terms, time-dependent changes in distal signalling, and constraints on some of the spatial variables. This model was again reimplemented as a single LimbNET model specification, which upon simulation successfully reproduces the published results: the generation of a five-digit striped pattern in the limb bud autopod (Fig. 4Bi), arising from a Turing patterning mechanism modulated by a time-dependent distal-to-proximal gradient of FGF. Secondly, we investigated the behaviour of the model under conditions more akin to a mesenchymal cell culture or micromass (Fig. 4Bii), by re-simulating the model specification in a square-shaped 2D domain. Through our model, we successfully reproduced experimentally observed results (Raspopovic et al, 2014): implanting virtual beads, soaked in either BMP or Wnt ligands, into virtual micromass culture results in local upregulation of sox9 in the case of BMP beads (Fig. 4Bii) and likewise in local inhibition and distal upregulation of sox9 expression for Wnt-soaked beads (Fig. 4Biii). Overall, LimbNET is able to successfully recapitulate previously published models and simulations, provided in a user-friendly web-based platform which allows users to explore, understand and modify the original models.

## Discussion

Understanding complex multi-scale biological phenomena will require computational modelling, but this will in turn require the sharing of models and simulations between a diverse community of theoreticians and experimentalists. It is important that predictive computer modelling facilitates collaboration and the shared exploration of new ideas and hypotheses. LimbNET is a first step towards this vision, specifically oriented towards the paradigmatic model system of limb development.

We believe that an open web-based simulation platform linked to a core common data framework provides the ideal scenario for getting scientists on the same page—ensuring both data and ideas can be compared in an objective way. Sharing data per se (Fig. 5A) is only the first step. Prior to LimbNET two images of the same gene expression pattern from two different research groups may both be on the same website, and they may be comparable by eye. It is also true that the conclusions we wish to derive are often qualitative: "the mutant gene expression pattern was more proximal than the wild-type". However, although that type of useful statement is qualitative, it requires a quantitative comparison to prove whether it is true, going beyond pure digitisation/quantification to objective comparison. While data and its analysis may often be used to *describe* a system, mathematical models represent hypotheses *explaining* the system, with the symbiosis between data and models cultivating new insights. By coupling our data framework to a computational modelling platform, we hope to help researchers more easily explain complex patterning data via dynamical modelling (Fig. 5B). Very simple causal relationships may not require computer modelling to explain or understand them. For example, the question "which upstream Transcription Factor activates gene X?" may be answered directly by a purely experimental approach, such as Chip-Seq (Osterwalder et al, 2014). Indeed, a computational dynamical model cannot answer such a direct empirical question. However, if we want to understand how a *network of genes* is responsible for shaping dynamic spatial expression patterns over time, this is too complex to understand purely by mental thought processes—a true understanding can only be gained through model-based analysis of the system (Balaskas et al, 2012; Jaeger et al, 2004a).

Computer modelling is necessary to understand the dynamic and emergent behaviour of complex networks, but empirical data from experimental perturbations are of course essential for building, verifying or disputing model predictions, e.g. knock-outs, ectopic expression or beads experiments (Fig. 5C). A single platform that includes both digitised experimental data and also model predictions therefore greatly facilitates comparison of hypotheses with quantitative data (both for the wild-type case and experimental perturbations). Crucially, this may lead to invalidation of model predictions under certain conditions (direct biochemical perturbations, mutations) thus offering further insights into hidden mechanisms, and the opportunity to (re)build existing work and foundations laid by prior modelling efforts (Fig. 5D).

LimbNET's tight integration between models/simulations and digitised data also allows researchers to automate parameter fitting of proposed regulatory networks to experimental observations (Fig. 5E). Given a topology of regulatory network interactions, formalised as a system of PDEs, the user may choose certain parameters to be automatically optimised, to give the closest result to the empirical data. Parameter optimisation operates by defining an objective function that attempts to minimise the discrepancy between simulation outputs and digitised data, in order to attempt

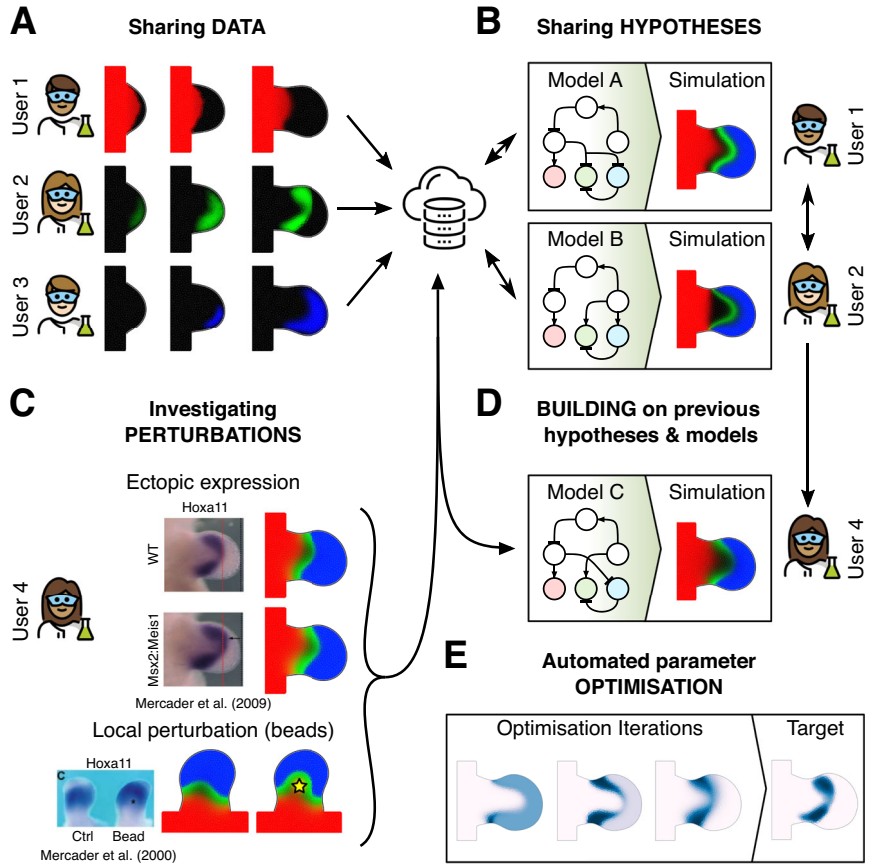

**Figure 5. Hypothesis-driven open science.**

The most important novelty of LimbNET is the ability to share ideas/hypotheses plus the simulation tools to run them, rather than being purely a "database". Nevertheless, its power lies in being data-driven, and it is indeed important that users are able to share spatiotemporal patterning data via a central data repository (**A**). In this illustrative example, three users have uploaded three separately acquired spatiotemporally varying gene expression patterns acquired from imaging data; Meis (red); Hoxa11 (green); Hoxd13 (blue). In addition to the expression data, LimbNET also provides for the sharing of hypotheses in the form of mathematical models (**B**). Users may upload models of regulatory networks specifying some form of spatiotemporally varying GRN dynamics, and simulate them. For example, two users observe the data uploaded in (**A**) and devise two different hypotheses for the underlying behaviour, formalised as two different regulatory network models ("Model A" and "Model B"). They upload the equations to LimbNET and are able to simulate them and compare the predictions generated by both interaction networks. Given a regulatory network, and corresponding mathematical model, users may compare the model output to further experiments (**C**), for example, perturbations of the original experimental system (indicated images/panels reproduced from (Uzkudun et al, 2015)). They—or indeed any other collaborators with access to LimbNET—may then refine or change the previous model(s) (**D**), essentially building on previous biological hypotheses and refining them to take into account new observations. (**E**) Automated parameter optimisation is also an important tool within LimbNET, and is accessible both through the web client, and the API.

to fit a specific network model to a concrete set of observations (Uzkudun et al, 2015; Lobo and Levin, 2015; Mousavi and Lobo, 2024).

We believe that LimbNET offers a number of advantages over existing modelling platforms. Users do not need to perform complex installation procedures—likewise, developers should not need to support many different Operating Systems—and the speed of simulation does not depend on users' own hardware. A centralised resource (for both data and modelling) helps to ensure internal consistency of data (images, digitisations, models and simulations). In addition, this type of service facilitates sharing and collaboration (Byrne et al, 2010). Users can make data private, if need be, and conversely published models can be made public.

Reproducibility in biological modelling is a well-known challenge (Tiwari et al, 2021), where predominant issues include missing parameter values, initial conditions or inconsistent model structure.

LimbNET dramatically reduces these problems. Although the backend (simulation engine) is remote/centralised, a model's equations, parameters, etc., are all explicitly specified and open. A visiting researcher can immediately simulate a published result and verify it themselves. Crucially, the visitor can also interrogate and challenge published models through their own perturbations, modifying parameters and extending the existing equations.

Two features of LimbNET may be seen as limitations, but in fact, we believe they strengthen the value of this platform to the community. Firstly, we wish the system to be both sophisticated in what it can achieve (a tight integration of both data analysis and predictive simulations) while also being very accessible, user-friendly and easy for non-specialists to use. To achieve this balance, LimbNET is specifically focused on just one developmental organ (currently geometries are provided for patterning of two species: the mouse limb bud, and the catshark fin bud) and on just one aspect of its

development (the control of gene expression patterns by GRNs). More generalised software such as Chaste (Mirams et al, 2013) and Morpheus (Starruß et al, 2014) represent more general-purpose tools, configurable to simulate a much wider range of developmental systems and phenomena. However, the price for greater generality is a higher energy barrier to non-specialist users. The goal for LimbNET is to focus on a particular community (limb development), and to make data-driven modelling as accessible as possible. Notwithstanding this goal, the LimbNET platform could indeed model a wide range of tissues and organs, if they can be represented by new 2D growing meshes. This is already supported by LimbNET's internal data structures and simulator architecture, since its underlying framework is capable of simulating reaction–diffusion systems on arbitrary 2D geometry, including growth and movement.

Secondly, while LimbNET's pre-defined wild-type tissue movements may at first appear limiting in terms of the limb developmental phenotypes that can be modelled, the current focus of LimbNET is to address the complexities of pattern formation by GRNs per se. A strong focus of the limb development community has been how gene regulatory interactions control the molecular patterning of genes, rather than the physical morphogenesis, and LimbNET thus aims to be valuable to this core interest of the field. In the near future, we plan to add extra mesh trajectories that will represent different degrees of growth impairment, or excessive growth (such as the *Xt* mutant (Johnson, 1967), a mutation in the *Gli3* gene (Schimmang et al, 1992)). Nevertheless, it is important to note that a large proportion of reported genetic experiments (both knock-outs and ectopic expression) reveal shifts in gene expression patterns without significant growth defects. The majority of these have not yet been explained through GRN models of spatiotemporal signalling, something which will now be possible.

In conclusion, LimbNET is itself an experiment in collective and open science. It is primarily aimed at the extensive experimental community, and it will be maintained, supported and further developed by the authors (although we are very happy for other groups to co-develop if the motivation arises). Although it is designed to allow any researcher to develop and explore their own models, we plan to steadily increase the number of models available over the years, always making them openly accessible to the scientific community. We will also increase the amount of gene expression data, both from our own data and from the literature, and help others to do the same. We hope that this tool evolves into a valuable, open forum in which the core currency of scientific debate—causal hypotheses—can be discussed and shared within a common and transparent frame of reference.

## Methods

### Reagents and tools table

| Reagent/ resource | Reference or source | Identifier or catalogue number |
|---|---|---|
| **Experimental models** | | |
| N/A | | |
| **Recombinant DNA** | | |
| N/A | | |
| **Antibodies** | | |

| Reagent/ resource | Reference or source | Identifier or catalogue number |
|---|---|---|
| N/A | | |
| **Oligonucleotides and other sequence-based reagents** | | |
| N/A | | |
| **Chemicals, enzymes and other reagents** | | |
| N/A | | |
| **Software** | | |
| Flask 3 | https:// flask.palletsprojects.com | |
| Vue.js 3 | https://vuejs.org | |
| PostgreSQL 14 | https://www.postgresql.org | |
| Redis 7 | https://redis.io/ | |
| RQ | https://python-rq.org/ | |
| **Other** | | |
| N/A | | |

## Data availability

The LimbNET application presented in this study can be accessed at https://limbnet.embl.es. This study includes no data deposited in external repositories. Source code is available at https://git.embl.org/grp-simbiont/limbnet/limbnet.

The source data of this paper are collected in the following database record: biostudies:S-SCDT-10_1038-S44320-025-00128-y.

## Peer review information

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

## Acknowledgements

The authors thank members of the Sharpe Lab for useful discussions and feedback, especially Philipp Germann, Xavier Diego, Marco Musy, Heura Cardona, June JuYeon Han, Lau Avinyo and James Cotterell. We would like to thank EMBL's IT hardware and support staff, in particular Thomas von Kiedrowski and BioIT. This project was supported by funding from EMBL, a European Research Council (ERC) Advanced grant (SIMBIONT, project no. 670555), and Spanish National Plan for Scientific and Technical Research and Innovation (Plan Estatal) Grants (PID2019-110868GB-I00 and PID2022-140399NB-I00).

## Author contributions

**Antoni Matyjaszkiewicz**: Data curation; Software; Validation; Investigation; Visualisation; Methodology; Writing—original draft; Writing—review and editing. **James Sharpe**: Conceptualisation; Resources; Supervision; Funding acquisition; Methodology; Project administration; Writing—review and editing.

Source data underlying figure panels in this paper may have individual authorship assigned. Where available, figure panel/source data authorship is listed in the following database record: biostudies:S-SCDT-10_1038-S44320-025-00128-y.

## Funding

## Disclosure and competing interests statement

James Sharpe is a member of the Advisory Editorial Board of Molecular Systems Biology. This has no bearing on the editorial consideration of this article for publication.

