## [Peer Review File · Molecular Systems Biology]

LimbNET: collaborative platform for simulating spatial patterns of gene networks in limb development

Antoni Matyjaskiewicz and James Sharpe

Corresponding author(s): Antoni Matyjaskiewicz (antoni.matyjaskiewicz@embl.es) , James Sharpe (james.sharpe@embl.es)

Review Timeline:

Submission Date:	7th Aug 24
Editorial Decision:	9th Aug 24
Appeal Received:	14th Aug 24
Editorial Decision:	1st Oct 24
Revision Received:	2nd Apr 25
Editorial Decision:	23rd Apr 25
Revision Received:	5th Jun 25
Accepted:	6th Jun 25

Editor: Jingyi Hou

Transaction Report:

9th Aug 2024

Manuscript Number: MSB-2024-12527

Dear Dr. Matyjaszkiewicz,

Thank you for submitting your manuscript "LimbNET: collaborative platform for simulating spatial patterns of gene networks in limb development" to Molecular Systems Biology.

Our editorial team performs an initial editorial assessment of all submissions. This process ensures that we handle all submitted manuscripts in a timely manner and that we send to reviewers only those studies that appear as sufficiently compelling candidates for publication in our journal.

We have now discussed your manuscript within our editorial team. I am afraid that we have decided to not send it out for peer review, as it does not seem well suited for publication in Molecular Systems Biology.

I apologise for not bringing better news, but I hope that this early decision will allow you to proceed without further delays.

Yours sincerely,

Jingyi Hou
on behalf of the Molecular Systems Biology editorial team

Jingyi Hou, PhD
Scientific Editor
Molecular Systems Biology

** As a service to authors, EMBO Press offers the possibility to directly transfer declined manuscripts to another EMBO Press title or to the open access journal Life Science Alliance launched in partnership between EMBO Press, Rockefeller University Press and Cold Spring Harbor Laboratory Press. The full manuscript and if applicable, reviewers' reports, are automatically sent to the receiving journal to allow for fast handling and a prompt decision on your manuscript. For more details of this service, and to transfer your manuscript please click on Link Not Available. **

The authors appealed the decision.

1st Oct 2024

Manuscript Number: MSB-2024-12527R-Q

Title: LimbNET: collaborative platform for simulating spatial patterns of gene networks in limb development

Author: Antoni Matyjaszkiewicz

James Sharpe

Dear Dr. Matyjaszkiewicz,

Thank you for submitting your work to Molecular Systems Biology. We have now heard back from the three reviewers who agreed to evaluate your manuscript. As you will see from the comments below, the reviewers find the manuscript to be of interest. They raise, however, several points, which should be addressed in a revision of this work.

The reviewers' recommendations are quite clear, and I see no need to reiterate their comments. Notably, Reviewer #3 pointed out that limbNET is currently somewhat limited and recommended incorporating additional models, genes, and mutant patterns to improve the platform's utility. These suggestions should be thoughtfully addressed where feasible.

All other issues need to be satisfactorily addressed as well. As you may already know, our editorial policy allows in principle a single round of major revision, and it is therefore essential to provide responses to the reviewers' comments that are as complete as possible. Please feel free to contact me in case you would like to discuss in further detail any of the issues raised by the reviewers.

On a more editorial level, we would ask you to address the following issues:

- Please provide a .docx formatted version of the manuscript text (including legends for main figures, EV figures and tables). Please make sure that the changes are highlighted to be clearly visible.
- Please provide individual production quality figure files as .eps, .tif, .jpg (one file per figure).
- Please provide a .docx formatted letter INCLUDING the reviewers' reports and your detailed point-by-point responses to their comments. As part of the EMBO Press transparent editorial process, the point-by-point response is part of the Review Process File (RPF), which will be published alongside your paper.
- Please note that all corresponding authors are required to supply an ORCID ID for their name upon submission of a revised manuscript.
- We replaced Supplementary Information with Expanded View (EV) Figures and Tables that are collapsible/expandable online (see examples in <http://msb.embopress.org/content/11/6/812>). A maximum of 5 EV Figures can be typeset. EV Figures should be cited as 'Figure EV1, Figure EV2' etc... in the text and their respective legends should be included in the main text after the legends of regular figures.

Additional Tables/Datasets should be labeled and referred to as Table EV1, Dataset EV1, etc. Legends have to be provided in a separate tab in case of .xls files. Alternatively, the legend can be supplied as a separate text file (README) and zipped together with the Table/Dataset file.

For the figures and tables that you do NOT wish to display as Expanded View figures, they should be bundled together with their legends in a single PDF file called *Appendix*, which should start with a short Table of Content. Each legend should be below the corresponding Figure/Table in the Appendix. Appendix figures and tables should be referred to in the main text as: "Appendix Figure S1, Appendix Figure S2, Appendix Table S1" etc. See detailed instructions regarding expanded view here: <https://www.embopress.org/page/journal/17444292/authorguide#expandedview>.

- Before submitting your revision, primary datasets (and computer code, where appropriate) produced in this study need to be deposited in an appropriate public database (see <http://msb.embopress.org/authorguide-dataavailability> <https://www.embopress.org/page/journal/17444292/authorguide#dataavailability>). Please remember to provide a reviewer password if the datasets are not yet public. The accession numbers and database should be listed in a formal "Data Availability" section (placed after Materials & Method) that follows the model below (see also <https://www.embopress.org/page/journal/17444292/authorguide#dataavailability>). Please note that the Data Availability Section is restricted to new primary data that are part of this study.

Data availability

- RNA-Seq data: Gene Expression Omnibus GSE46843 (<https://www.ncbi.nlm.nih.gov/geo/query/acc.cgi?acc=GSE46843>)

- [data type]: [name of the resource] [accession number/identifier/doi] ([URL or identifiers.org/DATABASE:ACCESSION])

-At EMBO Press we ask authors to provide source data for the main figures. Our source data coordinator will contact you to discuss which figure panels we would need source data for and will also provide you with helpful tips on how to upload and organize the files.

- Our journal encourages inclusion of *data citations in the reference list* to directly cite datasets that were re-used and obtained from public databases. Data citations in the article text are distinct from normal bibliographical citations and should directly link to the database records from which the data can be accessed. In the main text, data citations are formatted as follows: "Data ref: Smith et al, 2001". In the Reference list, data citations must be labeled with "[DATASET]". A data reference must provide the database name, accession number/identifiers and a resolvable link to the landing page from which the data can be accessed at the end of the reference. Further instructions are available at .

- We updated our journal's competing interests policy in January 2022 and request authors to consider both actual and perceived competing interests. Please review the policy <https://www.embopress.org/competing-interests> and update your competing interests if necessary. Please use the heading "Disclosure statement and competing interests".

- All Materials and Methods need to be described in the main text using our 'Structured Methods' format. According to this format, the Methods section includes a Reagents and Tools Table (listing key reagents, experimental models, software and relevant equipment and including their sources and relevant identifiers) followed by a Methods and Protocols section describing the methods, ideally using a step-by-step protocol format. The aim is to facilitate adoption of the methodologies across labs. Please download and fill our Reagents and Tools Table template (.docx), which you can find in our author guidelines: <https://www.embopress.org/page/journal/17444292/authorguide#structuredmethods>.

An example of a Method paper with Structured Methods can be found here: <https://www.embopress.org/doi/10.15252/msb.20178071>.

-Regarding data quantification:

Please ensure to specify the name of the statistical test used to generate error bars and P values, the number (n) of independent experiments (please specify technical or biological replicates) underlying each data point and the test used to calculate p-values in each figure legend. Discussion of statistical methodology can be reported in the materials and methods section, but figure legends should contain a basic description of n, P and the test applied.

Graphs must include a description of the bars and the error bars (s.d., s.e.m.). Please also include scale bars in all microscopy images.

- Please provide a "standfirst text" summarizing the study in one or two sentences (approximately 250 characters, including space), three to four "bullet points" highlighting the main findings and a "synopsis image" (550px width and 400-600 px height, PNG format) to highlight the paper on our homepage.

Here are a couple of examples:

<https://www.embopress.org/doi/10.15252/msb.20199356>

<https://www.embopress.org/doi/10.15252/msb.20209475>

<https://www.embopress.org/doi/10.15252/msb.209495>

When you resubmit your manuscript, please download our CHECKLIST (<https://www.embopress.org/pb-assets/embosite/EMBO%20Press%20Author%20Checklist-1642513524327.xlsx>) and include the completed form in your submission.

Please note that the Author Checklist will be published alongside the paper as part of the transparent process (<https://www.embopress.org/page/journal/17444292/authorguide#transparentprocess>).

If you feel you can satisfactorily deal with these points and those listed by the referees, you may wish to submit a revised version of your manuscript. Please attach a covering letter giving details of the way in which you have handled each of the points raised by the referees. A revised manuscript will be once again subject to review and you probably understand that we can give you no guarantee at this stage that the eventual outcome will be favorable.

I look forward to receiving your revised manuscript soon.

Sincerely,
Jingyi

We realize that it is difficult to revise to a specific deadline. In the interest of protecting the conceptual advance provided by the work, we recommend a revision within 3 months (30th Dec 2024). Please discuss the revision progress ahead of this time with the editor if you require more time to complete the revisions. Use the link below to submit your revision:

IMPORTANT: When you send your revision, we will require the following items:

1. the manuscript text in LaTeX, RTF or MS Word format
2. a letter with a detailed description of the changes made in response to the referees. Please specify clearly the exact places in the text (pages and paragraphs) where each change has been made in response to each specific comment given
3. three to four 'bullet points' highlighting the main findings of your study
4. a short 'blurb' text summarizing in two sentences the study (max. 250 characters)
5. a 'thumbnail image' (550px width and max 400px height, Illustrator, PowerPoint or jpeg format), which can be used as 'visual title' for the synopsis section of your paper.
6. Please include an author contributions statement after the Acknowledgements section (see <https://www.embopress.org/page/journal/17444292/authorguide>)
7. Please complete the CHECKLIST available at (<https://bit.ly/EMBOPressAuthorChecklist>). Please note that the Author Checklist will be published alongside the paper as part of the transparent process (<https://www.embopress.org/page/journal/17444292/authorguide#transparentprocess>).
8. When assembling figures, please refer to our figure preparation guideline in order to ensure proper formatting and readability in print as well as on screen:
<https://bit.ly/EMBOPressFigurePreparationGuideline>
See also figure legend guidelines: <https://www.embopress.org/page/journal/17444292/authorguide#figureformat>
9. Please note that corresponding authors are required to supply an ORCID ID for their name upon submission of a revised manuscript (EMBO Press signed a joint statement to encourage ORCID adoption). (<https://www.embopress.org/page/journal/17444292/authorguide#editorialprocess>)
Currently, our records indicate that the ORCID for your account is 0000-0002-8647-5773.

Link Not Available

10. At EMBO Press we ask authors to provide source data for the main manuscript figures. Our source data coordinator will contact you to discuss which figure panels we would need source data for and will also provide you with helpful tips on how to upload and organize the files.
11. Include a Reagents and Tools Table as part of the Methods section, which can be downloaded from our author guidelines (<https://www.embopress.org/page/journal/17444292/authorguide#structuredmethods>)

*** PLEASE NOTE *** As part of the EMBO Press transparent editorial process initiative (see our Editorial at <https://dx.doi.org/10.1038/msb.2010.72>), Molecular Systems Biology publishes online a Review Process File with each accepted manuscripts. This file will be published in conjunction with your paper and will include the anonymous referee reports, your point-by-point response and all pertinent correspondence relating to the manuscript. If you do NOT want this File to be published, please inform the editorial office at msb@embo.org within 14 days upon receipt of the present letter.

Reviewer #1:

Summary

This manuscript presents LimbNET, a new web tool for spatial and temporal morphological gene expression patterning visualization and simulation during mouse limb development. The two main components are the dataset of mapped gene

expression patterns in 2D limb reconstructions and the simulation of reaction-diffusion models that can generate such gene expression patterns. The interface is extremely user-friendly and allows users to upload new experimental data as well as rapidly create new models and simulations.

General remarks

LimbNET represents an ideal tool for limb development researchers to share gene expression data as well as mechanistic models. It has the potential to become a centralized repository of spatiotemporal experimental data for the community, as well as of mechanistic models that aims to explain such data. In summary, LimbNET is one of the most advanced repositories combining spatiotemporal data and spatial simulations and will be of interest for both experimentalists, theoreticians, and systems biologists.

Major points

1. One of the main novelties of LimbNET is the tight integration of experimental data and mechanistic models in a single platform. However, the current interface lacks the ability to compare the constructed models (hypothesis) with the experimental data that it tries to explain. I recommend the ability to visualize side-by-side the experimental data and the simulations for researchers to be able to evaluate the quality of the model. In addition, a numerical or visual comparison (e.g., image subtraction) between experimental and simulated data in the interface would be very useful.
2. Although the interface to define models is user-friendly, it lacks fundamental functionality for importing or exporting the model equations in common formats such as MathML (or SBML). That will extend the interoperability of this platform with other tools.
3. It is unclear if there are plans to curate more experimental data from the literature or if the authors envision individual labs to submit their data (and models) to the repository.
4. The web tool requires a user and a password to access it. It is not clear if the tool will be freely available after the manuscript is published. If that is the case, are there are plans to have a curation team to validate the images and models published in the database by users?
5. One of the main advantages of a web tool is the sharing of information (such as models) with unique URLs that can be emailed and included in manuscripts, and accessed without any login requirements. It is not clear if that functionality is planned but it would be essential for the widespread dissemination of the models (and potentially datasets) created with the tool.

Minor points

6. It would be useful to be able to select the stage of an image manually. Currently it seems it cannot be corrected once the algorithm fits it to one.
7. In addition to uploading images individually, an option to bulk upload and processing datasets (zip files?) would be ideal.
8. The title should include "mouse" limb development.
9. The channel assignments of a simulation are not saved, which makes going back and forth between changing the model and simulating it tedious.
10. It would be useful to cache the simulations, so new users do not need to repeat simulations already performed. This would be especially useful with published models.
11. Simulation errors show NaN values. Instead, a warning or report would be more useful.

Reviewer #2:

Matyjaszkiewicz and Sharpe present LimbNET, as a web-based platform for simulating and visualizing spatiotemporal gene expression patterns in limb development. The platform is designed to allow sharing of data, model building and simulation. The platform would act as a repository for datasets for limb development gene expression patterns and would also allow experimentalists to test theoretical models by simulations. The manuscript provides a comprehensive description of LimbNET's features, implementation details, and case studies, demonstrating its potential for advancing our understanding of limb

development. It allows users to upload in situ/fluorescent in situ images which can then be digitized to compare with simulations. Partial differential equations based models parameters can be added to LimbNET and the backend takes care of simulation infrastructure. Clever prevention of redrawing of mesh prevents excess computation loads for simulations. The collaborative nature of sharing data and models holds potential for newer models and hypothesis testing. The manuscript is well written with a defined focus and clear discussions on the limitations and future directions.

Points:

1. LimbNET focuses on limb development. However, the current version majorly focuses on the mouse limb bud. While the authors acknowledge some limitations of LimbNET, predetermined tissue movements, a more thorough discussion regarding data from other species and the future plans regarding the same would significantly enhance the manuscript. Opening of APIs for other species could be implemented for future use. This would increase LimbNET's applicability and contribute to comparative developmental biology research.
2. Authors provide the ability to upload imaging data for gene expression pattern. However, without specifications regarding the image file format, if any preprocessing of the images is required or other guidelines, it might be challenging to ensure data quality and reproducibility.
3. It would be beneficial if the authors addressed the long-term maintenance of the platform. Neglecting this aspect could lead to data loss, platform instability, or decreased user trust, hindering the platform's potential impact.
4. Provision of API will help power users integrate LimbNET in scripts/workflows. Including details and examples on API usage in the documentation/manuscript would make it more appealing for power users and demonstrate the full capabilities of the platform.

Overall, this is a well-written manuscript with a clear focus on a valuable tool for the field of developmental biology. Addressing the points mentioned above would further enhance the manuscript's value and impact.

Reviewer #3:

Matyjaszkiewicz et al., propose a platform for gene patterns simulation that, for the moment, relies on already published models of limb proximo distal patterning and digit formation. It is intended for limb development researchers and allows not only to simulate and test their hypothesis, but also to share data (gene expression images and simulations). The platform includes an in situ hybridization database in which researchers can upload their own images that can be staged and 'digitised' for further implementation into the modeling pipeline. The interface is easy to use and it does not require programming skills, nor special software. The goal of the authors is to make modelling accessible to all researchers and to bring together all limb field knowledge in a standardized manner.

My main concern about limbNET is that currently is rather limited because it only allows simulations based on two published models (Uzkudun et al., 2015 and Rapopovic et al., 2014). Even though it is proposed to make modelling accessible for all researchers, the update and development of the new models relies on experts. This limitation might make limbNET not a very useful tool for the whole limb community as it does not cover many other aspects of limb research. Are the authors planning to develop models of dorso-ventral and antero-posterior patterning for instance? Also, although already mentioned by the authors in the text, it would be a great improvement for the platform to add mutant gene patterns. There are a lot of limb phenotypes with in situ hybridization data published which, added to the models, would strengthen the GRN complexity and understanding. Do the authors consider a way to maintain it updated with new published data and new gene interactions? In relation to this another concern is that models are limited to normal growth (already discussed by authors) and few gene sets. Digit formation model is limited to Sox9, Bmp and Wnt and PD patterning model relies just in Meis, Hoxa11, Hoxa13, RA and FGF signaling. This simplification does not reflect the complexity of gene regulatory networks occurring during development.

Overall, although I found limbNET very limited at this point, I am positive about it and I recommend the manuscript for publication. It holds potential to become a powerful tool in the future, once it is reinforced with more models, more genes, mutants' data ...as well as the integration of several models within the same simulation, reflecting the actual developing limb's gene regulatory network scenario. For this.

Minor points:

- For most uploaded images present in the platform it is not stated whether it is an image of a FL or a HL

-Is there is any quality control for the images that are uploaded in the platform? I found several images with poor quality. It would be preferable to have less images with higher quality rather than a lot of images displaying undistinguishable patterns.

-It is not clear to me how to integrate new image data in a simulation once they are 'digitised'. Can any user do it? Or is the expert (the authors) the ones that need to combine all the images of a given gene in order to include it in the modeling pipeline?

-Fig.4 iii) It is my understanding that AER removal in the model is mimicked by FGF deletion. In this case I would expect the same patterns as the ones in FGF mutants. But this it is not the case: Hoxa11 expression in Fgf4,8,9KO and Fgf8,9KO is not

distal, as shown in the simulation, but restricted to the intermediate normal region (Mariani et al., 2008 SuppFig1). Could the author comment on this?

As a general comment to the editor and all reviewers, we have very much appreciated the feedback and positive comments.

Our response has been a little delayed, because we decided to implement into the code of LimbNET as many of the suggestions as possible. Major upgrades we have added include:

- A side-by-side double-viewer mode, so researchers can (for example) compare the real data with the simulated results.
- An image subtraction mode, as a direct way to visualise the differences between any 2 patterns (differences between simulated and real patterns, or 2 different simulated patterns, or 2 real gene expression patterns).
- A model export function, which can export the model equations as JSON or SBML.
- Improved warning messages and feedback (where there are simulation problems such as NaNs, etc.).
- The addition of a new model for the Sox9 patterning of a different species - the shark fin bud. This model was previously published by our lab (Onimaru et al. 2016), therefore strengthens the value of LimbNET for providing published models to the community
- Preserving the settings of channels when going back and forth between editor and simulator.
- Filters for information about whether the limb is forelimb or hindlimb in the image/staging listings.
- A new section of the interface to control parameter optimization runs.

Instructions of how to use these new features are being added into the online documentation.

We have also added a new paragraph into the discussion of the paper to explain many of the questions raised by the reviewers.

Reviewer #1:

Summary

This manuscript presents LimbNET, a new web tool for spatial and temporal morphological gene expression patterning visualization and simulation during mouse limb development. The two main components are the dataset of mapped gene expression patterns in 2D limb reconstructions and the simulation of reaction-diffusion models that can generate such gene expression patterns. The interface is extremely user-friendly and allows users to upload new experimental data as well as rapidly create new models and simulations.

General remarks

LimbNET represents an ideal tool for limb development researchers to share gene expression data as well as mechanistic models. It has the potential to become a centralized repository of spatiotemporal experimental data for the community, as well as of mechanistic models that aims to explain such data. In summary, LimbNET is one of the most advanced repositories combining spatiotemporal data and spatial simulations and will be of interest for both experimentalists, theoreticians, and systems biologists.

We very much appreciate the reviewer's positive comments.

Major points

1. One of the main novelties of LimbNET is the tight integration of experimental data and mechanistic models in a single platform. However, the current interface lacks the ability to compare the constructed models (hypothesis) with the experimental data that it tries to explain. I recommend the ability to visualize side-by-side the experimental data and the simulations for researchers to be able to evaluate the quality of the model. In addition, a numerical or visual comparison (e.g., image subtraction) between experimental and simulated data in the interface would be very useful.

We thank the reviewer for this suggestion. We agree that this is a good idea, and so we have directly implemented it.

In the simulation view, the mode can now be switched to a side-by-side viewer with two panels, each panel with three R/G/B channels that can be populated with digitised experimental data and/or simulation outputs. We implemented this with a single coordinated time-slider, so that both views are always synchronised. Here are 2 snapshots of this new feature - a younger and older time-point - in each case comparing simulated patterns from a model (left) with the real digitised data on the right.

We also agree that the idea of an image-subtraction mode is very appealing, and so we have implemented that as well. In this mode, 2 channels exist (rather than the usual 3 for RGB) and the image displayed shows the pattern of the difference values (pink are negative values, and green are positive values) as shown below:

2. Although the interface to define models is user-friendly, it lacks fundamental functionality for importing or exporting the model equations in common formats such as MathML (or SBML). That will extend the interoperability of this platform with other tools.

We agree with the reviewer and acknowledge the value of interoperability with other tools. We have therefore now implemented two model exporters, available within the model editor interface, JSON and SBML:

- The JSON exporter allows the user to download a textual representation of LimbNET's internal model representation object as a JSON file including all equations, parameters and metadata.
- The SBML exporter produces a file listing the differential equations in SBML format. The exporter produces an SBML 3.2 compatible XML file, with the model's internal

representation converted to the closest equivalent SBML fields. For example, all reactants and variables are represented by SBML species.

One of the main goals of our modelling framework is to explore simulations that are not only temporal, but crucially, that are spatial. Indeed, explaining how spatial patterns arise is the primary objective. In this respect SBML has a clear limitation for representing our models, as although it can represent topological compartmentalisation, it does not include any explicit concept of space. For this reason some essential features of LimbNET's models, such as spatial prepatterns, cannot be encoded in a meaningful way within SBML. Nevertheless, we envisage that simply exporting the differential equations in a standardised format could be useful in certain circumstances, and therefore have provided the exporter.

The relevant menu with the new functionality can be found in the model editor (in the top right):

As an example of the new SBML functionality, we download an XML file containing the generated SBML specification for one model, and import it in the popular stochastic/ODE modelling package COPASI:

The equations, and relevant parameters of the model, are all correctly imported and parsed:

The fact that LimbNET requires dynamically-changing spatial domains which are represented as an interconnected series of triangular meshes (as well as the spatial prepatterns mentioned above) indicates why we have not implemented a model reader. We are not aware of any standardised model format which can include this information.

We must stress of course, the goal of LimbNET is that users can indeed save and load models into the platform from a database of LimbNET models, which is itself incorporated into the platform. Through the ability to make any model private or public, users can also openly share their models with any other LimbNET user. We will strive to make this environment open and user-friendly enough that there are no impediments in sharing, examining and testing all models.

3. It is unclear if there are plans to curate more experimental data from the literature or if the authors envision individual labs to submit their data (and models) to the repository.

We plan to pursue a variety of ways to populate the repository, including the two options mentioned above. We are in the process of curating more data, and more hypotheses from the literature. We also wish to encourage other labs to build and submit their own models. This can be either creating a model from scratch, or the easier approach of modifying an existing model already in the database. In addition to these options, we are starting to collaborate with a few other groups to help them convert their hypotheses into models. Overall, the project is itself an experiment for the community as a whole. We will put as much effort as we can into helping the whole community to use this platform, and improving it based on feedback. We have added some more text to the discussion, to make this general philosophy clearer.

4. The web tool requires a user and a password to access it. It is not clear if the tool will be freely available after the manuscript is published. If that is the case, are there plans to have a curation team to validate the images and models published in the database by users?

Access to LimbNET does require a username and password, but we will freely give access to anyone who requests it. There are some non-trivial security implications, associated with a completely open platform, such as (for example) upload of unsuitable images or data, as

well as vandalism. As a very small team, it would be beyond our ability to moderate the platform if it were completely open, so requiring a username is one way of protecting the system from possible malicious behaviour. Having contact information for users also allows us to contact them if there is a technical problem or a bug that needs to be fixed, and to inform if they should run their simulations again.

As mentioned above, this project is an experiment for the community. When users are exploring data and models in their private space we will not curate or validate anything - it is important for users to have the freedom to explore their own ideas freely. But when they make their data or models public we will check them (transparently in contact with the user) to avoid any unintentional mistakes on their part.

5. One of the main advantages of a web tool is the sharing of information (such as models) with unique URLs that can be emailed and included in manuscripts, and accessed without any login requirements. It is not clear if that functionality is planned but it would be essential for the widespread dissemination of the models (and potentially datasets) created with the tool.

We thank the reviewer for their excellent suggestion - we agree that this would be a brilliant way to make models completely public, for example in manuscripts etc. Indeed, already now LimbNET users can share models with each other by simply sharing the URL of the model.

We do plan to upgrade the system so that certain designated models could have a fully public URL, not requiring a login and password. However, this requires quite a significant re-programming of the code - treating "published" models as static assets with fixed IDs that can no longer be modified. Due to manpower constraints this is on the to do list, but will be done later.

Minor points

6. It would be useful to be able to select the stage of an image manually. Currently it seems it cannot be corrected once the algorithm fits it to one.

Yes, we agree that this would be useful in some scenarios. We are currently implementing this, and a mechanism for morphing from one stage to nearby ones (plus/minus a few hours).

7. In addition to uploading images individually, an option to bulk upload and processing datasets (zip files?) would be ideal.

We agree that this would be a useful feature, especially once the platform gains momentum and there is a greater need for uploading groups of related images. Although we have not been able to implement it yet, the feature is on our roadmap. There are some potential security implications - sometimes nontrivial - arising from allowing more generic upload services and these must be taken into consideration if we want to enable upload of larger zip archives.

8. The title should include "mouse" limb development.

In response to other reviewers comments, we have now added a model from a different species, the catshark (Onimaru et al. (2016) Nature Communications). We would therefore like to retain the current title. :

(Top) Image from the original paper of Onimaru et al. (2016). (Bottom) Image of the same model now accessible to users within LimbNET.

9. The channel assignments of a simulation are not saved, which makes going back and forth between changing the model and simulating it tedious.

We agree - we have updated the simulation view to try to take this workflow into account. Now the simulation view's current channel assignments will be stored when leaving that view; and they are restored when returning to the view. This works for the process of editing a model going back-and-forth between the editor and the simulation view. There is potential for improvement in the future when we implement caching of simulation results.

10. It would be useful to cache the simulations, so new users do not need to repeat simulations already performed. This would be especially useful with published models.

We agree, caching the simulation outputs would make the model development process smoother, and at the same time would make comparisons between models - or the same model with different parameters, etc. - much easier. The feature is on our roadmap for implementation soon.

11. Simulation errors show NaN values. Instead, a warning or report would be more useful.

We agree. We have implemented a warning message from the server if any of the computed simulation results contains non-finite values (NaN, infinity, ...). In case any such values are

detected, a warning is displayed to the user upon completion of a simulation, including the compartment in which the error was detected.

Reviewer #2:

Matyjaszkiewicz and Sharpe present LimbNET, as a web-based platform for simulating and visualizing spatiotemporal gene expression patterns in limb development. The platform is designed to allow sharing of data, model building and simulation. The platform would act as a repository for datasets for limb development gene expression patterns and would also allow experimentalists to test theoretical models by simulations. The manuscript provides a comprehensive description of LimbNET's features, implementation details, and case studies, demonstrating its potential for advancing our understanding of limb development. It allows users to upload in situ/fluorescent in situ images which can then be digitized to compare with simulations. Partial differential equations based models parameters can be added to LimbNET and the backend takes care of simulation infrastructure. Clever prevention of redrawing of mesh prevents excess computation loads for simulations. The collaborative nature of sharing data and models holds potential for newer models and hypothesis testing. The manuscript is well written with a defined focus and clear discussions on the limitations and future directions.

We thank the reviewer for their kind comments and feedback on the project and the manuscript.

Points:

1. LimbNET focuses on limb development. However, the current version majorly focuses on the mouse limb bud. While the authors acknowledge some limitations of LimbNET, predetermined tissue movements, a more thorough discussion regarding data from other species and the future plans regarding the same would significantly enhance the manuscript. Opening of APIs for other species could be implemented for future use. This would increase LimbNET's applicability and contribute to comparative developmental biology research.

We agree with the reviewer that a more in depth discussion about the limitations and philosophy of the LimbNET project would improve the manuscript. We have therefore added extra text into the last 2 paragraphs of the discussion to cover these points.

Regarding other species, we have decided to address the reviewer's comment in the most constructive way possible - to add another model from a different species. Previously in the lab, we have made a model for the Sox9 patterning of the shark fin bud (Onimaru et al. (2016) Nature Communications). This was an extension of the BSW model (Raspopovic et al. (2014)) that was developed for mouse limb development. We have now added this shark model into LimbNET, as it:

- (a) Helps to show that LimbNET is not limited to 1 species.
- (b) Expands the collection of previously-published models that are now accessible to the public. Now LimbNET contains the models from 3 previous publications: Raspopovic et al. (2014), Uzkudun et al. (2015) and Onimaru et al. (2016).

(c) It is the most direct way to address the reviewer's comment (and those of the other reviewers as well).

Here are snapshots of 3 time-points from the shark model.

2. Authors provide the ability to upload imaging data for gene expression pattern. However, without specifications regarding the image file format, if any preprocessing of the images is required or other guidelines, it might be challenging to ensure data quality and reproducibility.

We have added the requested information in the UI. Now, the image upload dialog states the accepted image file formats, size and dimensions. We have also added a comment into the discussion about image quality control.

3. It would be beneficial if the authors addressed the long-term maintenance of the platform. Neglecting this aspect could lead to data loss, platform instability, or decreased user trust, hindering the platform's potential impact.

These issues are now also discussed in the new paragraph in the discussion of the paper. We explain that this is an experiment in itself, but that it is an absolutely central project to the Sharpe lab, and therefore we commit to keeping it running, maintaining it and improving it over the coming years.

4. Provision of API will help power users integrate LimbNET in scripts/workflows. Including details and examples on API usage in the documentation/manuscript would make it more appealing for power users and demonstrate the full capabilities of the platform.

This is a good point, and is something that we are working on. Documentation for the API usage is currently work in progress, as are examples.

Until now our intention has been to focus on features for non-power-users; the ideal target audience is experimentalists who may not necessarily want to use a programmatic API, at least initially. Indeed the core aim of LimbNET's first release is to open up the power of spatiotemporal modelling to all researchers in the limb development community, not just those seen as power users or theoreticians/modellers. Since the API has been changing as LimbNET evolves we have not released a stable API specification and documentation, however we will now be able to work on this.

Overall, this is a well-written manuscript with a clear focus on a valuable tool for the field of developmental biology. Addressing the points mentioned above would further enhance the manuscript's value and impact.

Reviewer #3:

Matyjaszkiewicz et al., propose a platform for gene patterns simulation that, for the moment, relies on already published models of limb proximo distal patterning and digit formation. It is intended for limb development researchers and allows not only to simulate and test their hypothesis, but also to share data (gene expression images and simulations). The platform includes an in situ hybridization database in which researchers can upload their own images that can be staged and 'digitised' for further implementation into the modeling pipeline. The interface is easy to use and it does not require programming skills, nor special software. The goal of the authors is to make modelling accessible to all researchers and to bring together all limb field knowledge in a standardized manner.

We thank the reviewer for their kind comments and feedback regarding the interface and the manuscript.

My main concern about limbNET is that currently is rather limited because it only allows simulations based on two published models (Uzkudun et al., 2015 and Rapopovic et al., 2014). Even though it is proposed to make modelling accessible for all researchers, the update and development of the new models relies on experts. This limitation might make limbNET not a very useful tool for the whole limb community as it does not cover many other aspects of limb research. Are the authors planning to develop models of dorso-ventral and antero-posterior patterning for instance?

We should first clarify that LimbNET is not limited to the two models of Uzkudun et al., 2015 and Raspopovic et al., 2014. These are indeed the only published models of mouse limb development that we have already uploaded into the model database. But the platform is a community experiment in its own right. We plan to pursue a variety of ways to populate the repository. We are in the process of curating more data, and more hypotheses from the

literature. We also wish to encourage other labs to build and submit their own models. This can be either creating a model from scratch, or the easier approach of modifying an existing model already in the database. In addition to these options, we are starting to collaborate with a few other groups to help them convert their hypotheses into models. We will put as much effort as we can into helping the whole community to use this platform, and improving it based on feedback. We have added some more text to the discussion, to make this general philosophy clearer.

We are happy to report here in the rebuttal, that we have now added another model, which is of a different species. Another paper from the Sharpe lab published a model of skeletal patterning in the catshark fin - Onimaru et al. (2016)

We should also clarify:

- Regarding antero-posterior patterning, indeed LimbNET can model this phenomenon, and we are currently working on models that include Shh, Grem and other key factors.
- Regarding dorso-ventral patterning, LimbNET is only a 2D platform, and will therefore not be able to model this aspect.

Also, although already mentioned by the authors in the text, it would be a great improvement for the platform to add mutant gene patterns. There are a lot of limb phenotypes with in situ hybridization data published which, added to the models, would strengthen the GRN complexity and understanding.

We totally agree - digitising mutant patterns is an important next step. For mutants which do not display growth defects this will be easy. For mutants with growth alterations (such as Gli3, Shh, or Fgf8) we will also create altered morphomovies.

Do the authors consider a way to maintain it updated with new published data and new gene interactions?

Indeed, this is a very important goal of the project, which we have explained further in the new paragraph in the discussion of the paper (as also explained in answer to some of the questions above).

In relation to this another concern is that models are limited to normal growth (already discussed by authors) and few gene sets.

We completely agree that this is a limitation of the platform in its current form. We are currently working on incorporating a diverse set of new gene expression patterns in the wild type mouse limb bud using advanced reconstruction methods to generate smooth time series from snapshots such as the exact images and digitisations that are already in our database (see Aviñó-Esteban et al. (2025) Development - <https://doi.org/10.1242/dev.204313>). This method is also applicable to more arbitrary domains (shapes, and their spatiotemporal evolution) and so could be extended and used for data from mutants, and potentially other organisms and/or organs in the future.

Regarding the comment on “normal growth”, in future we would like to put data from some growth mutants into the system (for example the extra growth of Gli3 mutants, and the reduced growth of Shh or Fgf8 mutants). Indeed, this would be the first step in extending the library of domains from what we currently have, before even considering other species or organs. However, for the time being we believe that the spatiotemporal growth model implemented in LimbNET already provides an extremely strong basis for generating interesting and useful models of patterning processes in the limb bud. During the stages of growth implemented in the current model (mouse e10:08 to e12:08) a large number of existing mutants do show changes in gene expression patterns, without great changes in the shape of the limb bud. Therefore the platform can already be used to study processes underlying signalling and patterning in a large subset of existing mutants, before having to consider changes in shape and growth.

Digit formation model is limited to Sox9, Bmp and Wnt and PD patterning model relies just in Meis, Hoxa11, Hoxa13, RA and FGF signaling. This simplification does not reflect the complexity of gene regulatory networks occurring during development.

We agree that real gene regulatory networks during development are more complex. The first goal of our LimbNET platform is to work with the state-of-the-art GRN models that already exist in the field, and to provide easy access to the rest of the community. The second goal is that this user-friendly platform will indeed help the community to improve the models of this complex process, to gradually and collectively become more and more accurate, which will presumably require the inclusion of more genes into the GRNs.

Overall, although I found limbNET very limited at this point, I am positive about it and I recommend the manuscript for publication. It holds potential to become a powerful tool in the future, once it is reinforced with more models, more genes, mutants' data ...as well as the integration of several models within the same simulation, reflecting the actual developing limb's gene regulatory network scenario. For this.

We thank the reviewer again for their useful comments and feedback on the platform. Our goal is to extend the platform with much more data, and models, and make it a useful resource for all scientists in the limb development community. As far as the integration of several models goes, we also mentioned this in the previous point, and we hope that this is something that will quickly come as the platform gains momentum.

Minor points:

- For most uploaded images present in the platform it is not stated whether it is an image of a FL or a HL

Yes, this was not something that was previously shown in the interface. We have now updated the interface to show whether an image is of a FL/HL, should a user desire to do so. For all of the existing images - where we know this - we have updated the image to indicate the F/H category.

-Is there is any quality control for the images that are uploaded in the platform? I found several images with poor quality. It would be preferable to have less images with higher quality rather than a lot of images displaying undistinguishable patterns.

Yes, we agree with the reviewer, we would prefer to have fewer images and of better quality - at least as a showcase/exemplar. There is an ongoing process of curation for images that are made public, that will become more important as the platform grows.

However many images that are not necessarily of prime quality are also useful data. For example an image that - as an individual data point - is not of especially high quality may nonetheless be useful in aggregate for further analysis. A specific case is generating reconstructions of spatiotemporal trajectories that may incorporate multiple images per time point and multiple time points (Aviñó-Esteban et al. (2025) Development - <https://doi.org/10.1242/dev.204313>).

We agree that it would be useful to somehow distinguish between these cases (and others as they arise).

To this end we are implementing the idea of collections of data(sets), already partially covered by the concept of tags and categories. We hope that all data that is useful - whether it is of high quality or not - will be publicly visible, but that through the use of collections and tags we can curate, indicate and promote data that are of a specifically high standard, solving the problem stated by the reviewer.

-It is not clear to me how to integrate new image data in a simulation once they are 'digitised'. Can any user do it? Or is the expert (the authors) the ones that need to combine all the images of a given gene in order to include it in the modeling pipeline?

The reviewer is correct that this is an important feature still missing for general users. We are therefore actively working on providing this functionality, which will be available in the very near future.

-Fig.4 iii) It is my understanding that AER removal in the model is mimicked by FGF deletion. In this case I would expect the same patterns as the ones in FGF mutants. But this it is not the case: Hoxa11 expression in Fgf4,8,9KO and Fgf8,9KO is not distal, as shown in the simulation, but restricted to the intermediate normal region (Mariani et al., 2008 SuppFig1). Could the author comment on this?

The modelling result referred to here, was in fact published in our earlier paper Uzkudun et al. (2015). That paper referred to an experimental result which was reported in Vargesson et al. (2001), which we reproduce here - in which Hoxa11 expression can still be seen right up to the distal-most tissue.

Indeed the Hoxa11 pattern from the Supplementary Fig1 of Mariani et a. (2008) is not fully distal. We would suggest this is evidence that the genotype, which was heterozygous for Fgf9 (Fgf8;Fgf4-DKO, Fgf9^{+/-}), still retained enough FGF signalling to keep Hoxa11 expression away from the very distal-most cells. By contrast the AER removal experiment of Vargesson et al. clearly removes all possible sources of FGF signalling.

23rd Apr 2025

Manuscript Number: MSB-2024-12527RR

Title: LimbNET: collaborative platform for simulating spatial patterns of gene networks in limb development

Author: Antoni Matyjaszkiewicz

James Sharpe

Dear Dr Matyjaszkiewicz,

Thank you for sending us your revised manuscript. We have now received feedback from the three reviewers who evaluated your study. As you will see below, the reviewers are overall satisfied with the performed revisions. Before we can formally accept the manuscript for publication, we would ask you to address some remaining minor issues listed below:

1. Please provide up to five key words.
 2. Please remove all figures from the main manuscript file. Figures should be uploaded separately as individual high-resolution files. Figure legends should remain in the manuscript and be placed after the References section.
 3. Ensure that the email addresses of all corresponding authors are included on the manuscript's title page.
 4. Remove "Note to reviewers".
 5. Data availability: since this study does not generate large-scale datasets, please also include the following sentence in this section- "This study includes no data deposited in external repositories".
 6. Please include the following sentence under the "Disclosure and competing interests statement" : "James Sharpe is a member of the Advisory Editorial Board of Molecular Systems Biology. This has no bearing on the editorial consideration of this article for publication." The entire section should be positioned after Acknowledgement.
 7. Please download and fill our Reagents and Tools Table template (.docx), which you can find in our author guidelines: <https://www.embopress.org/page/journal/17444292/authorguide#structuredmethods>.
- When submitting your revised manuscript, please do not include the Reagents and Tools Table in the Methods section of the manuscript but upload it as a separate file choosing the file type "Reagent Table".
8. Figure reuse 1D and 5C: This manuscript contains images that were published in a previous manuscript. Mol Syst Biol (2015) 11: 815 <https://doi.org/10.15252/msb.20145882>. While the earlier manuscript is called out in the current manuscript, the specific reuse of these figures must be explicitly stated in the corresponding figure legends.
 9. Please ensure that all funding information is entered into the online submission system and that it is consistent with the information provided in the manuscript file.
 10. Please remove the Authors' Contribution section from the manuscript file.
 11. Please add a "Results" heading at the beginning of the results section.
 12. Author checklist: Please provide corresponding authors' names, journal and manuscript ID.

Click on the link below to submit your revised paper.

Sincerely,
Jingyi

Jingyi Hou, PhD
Senior Editor
Molecular Systems Biology

*** PLEASE NOTE *** As part of the EMBO Press transparent editorial process initiative (see our Editorial at <https://dx.doi.org/10.1038/msb.2010.72> , Molecular Systems Biology will publish online a Review Process File to accompany accepted manuscripts. When preparing your letter of response, please be aware that in the event of acceptance, your cover letter/point-by-point document will be included as part of this File, which will be available to the scientific community. More information about this initiative is available in our Instructions to Authors. If you have any questions about this initiative, please contact the editorial office (msb@embo.org).

Reviewer #1:

The revised manuscript has addressed all my concerns.

Reviewer #2:

While not all of the reviewers requests have been fulfilled in the revised version, the authors have done an effort in
-improving operability of the platform
-include an additional model organism

After reading and comparing the requests as well as seeing the authors reply letter it became clear to me that many of our request were going beyond a regular revision.

The manuscript as such is already an important milestone in comparative expression pattern analysis of the developing limb. Hopefully it will constitute a tool to further be expanded with additional data and be used by the community. I did also understand the limits of a fully open source platform and satisfied about the sharing options the authors provide.

Reviewer #3:

Authors have addressed the comments/questions posed. Although they are still improving LimbNET and plan to incorporate new data and models, if these goals are achieved, LimbNET will definitely become an interesting tool for the field. Therefore, I recommend the manuscript for publication.

All editorial and formatting issues were resolved by the authors.

6th Jun 2025

Manuscript number: MSB-2024-12527RRR

Title: LimbNET: collaborative platform for simulating spatial patterns of gene networks in limb development

Dear Antoni,

Thank you again for sending us your revised manuscript. We are now satisfied with the modifications made and I am pleased to inform you that your paper has been accepted for publication.

Sincerely,
Jingyi

Jingyi Hou, PhD
Senior Editor
Molecular Systems Biology
